# Inferring a spatial code of cell-cell interactions across a whole animal body

Erick Armingol[1,2], Abbas Ghaddar[3], Chintan J. Joshi[2], Hratch Baghdassarian[1,2], Isaac Shamie[1,2], Jason Chan[4], Hsuan-Lin Her[1], Samuel Berhanu[3], Anushka Dar[3], Fabiola Rodriguez-Armstrong[3], Olivia Yang[3], Eyleen J. O'Rourke[3,5]*, Nathan E. Lewis[2,6]*

1 Bioinformatics and Systems Biology Graduate Program, University of California, San Diego, La Jolla, California, United States of America, 2 Department of Pediatrics, University of California, San Diego, La Jolla, California, United States of America, 3 Department of Biology, University of Virginia, Charlottesville, Virginia, United States of America, 4 Poway High School, Poway, California, United States of America, 5 Department of Cell Biology, School of Medicine of University of Virginia, Charlottesville, Virginia, United States of America, 6 Department of Bioengineering, University of California, San Diego, La Jolla, California, United States of America

* ejo8b@virginia.edu (EJO); nlewisres@ucsd.edu (NEL)

**Data Availability Statement:** The single-cell RNA-seq dataset (GEO accession code GSE98561), the 3D digital atlas of C. elegans including cell annotations based on the cell types in the

## Abstract

Cell-cell interactions shape cellular function and ultimately organismal phenotype. Interacting cells can sense their mutual distance using combinations of ligand-receptor pairs, suggesting the existence of a spatial code, i.e., signals encoding spatial properties of cellular organization. However, this code driving and sustaining the spatial organization of cells remains to be elucidated. Here we present a computational framework to infer the spatial code underlying cell-cell interactions from the transcriptomes of the cell types across the whole body of a multicellular organism. As core of this framework, we introduce our tool *cell2cell*, which uses the coexpression of ligand-receptor pairs to compute the potential for intercellular interactions, and we test it across the *Caenorhabditis elegans*' body. Leveraging a 3D atlas of *C. elegans*' cells, we also implement a genetic algorithm to identify the ligand-receptor pairs most informative of the spatial organization of cells across the whole body. Validating the spatial code extracted with this strategy, the resulting intercellular distances are negatively correlated with the inferred cell-cell interactions. Furthermore, for selected cell-cell and ligand-receptor pairs, we experimentally confirm the communicatory behavior inferred with *cell2cell* and the genetic algorithm. Thus, our framework helps identify a code that predicts the spatial organization of cells across a whole-animal body.

## Author summary

Neighboring cells coordinate gene expression through cell-cell interactions, enabling proper functioning in multicellular organisms. Hence, intercellular interactions can be inferred from gene expression. We use this strategy to define a molecular code bearing spatial information of cell-cell interactions across a whole animal body. We develop a computational framework to infer the first cell-cell interaction network in *Caenorhabditis*

scRNAseq dataset (S6 Table), the manual curated list containing 245 ligand-receptor interactions (S1 Table), and the consensus list from the GA-selection containing 37 interactions (S3 Table) are available in a public Code Ocean capsule (https://doi.org/10.24433/CO.4688840.v2). All analyses performed in this work, their respective codes (implemented in Python and Jupyter Notebooks), all data, and instructions to use them are available in a public repository (https://github.com/LewisLabUCSD/Celegans-cell2cell). Reproducible runs of our analyses can be performed in a public Code Ocean capsule (https://doi.org/10.24433/CO.4688840.v2). Our open-source suite, cell2cell, is for inferring cell-cell interactions from bulk or single-cell RNA-seq data, using or not spatial information, and is available in a GitHub repository (https://github.com/earmingol/cell2cell).

**Funding:** EA is supported by the Chilean Agencia Nacional de Investigación y Desarrollo (ANID) through its scholarship program DOCTORADO BECAS CHILE/2018 - 72190270, the Fulbright Chile Commission, and the Siebel Scholar Foundation. This work was further supported by NIGMS grant R35 GM119850 to NEL, a Lilly Innovation Fellows Award to CJJ, Jefferson Foundation Award to AG, J Yang Foundation Fellowship to HLH, PEW Charitable Trust Award and a generous funding from the W. M. Keck Foundation to EJO. The funders had no role in study design, data collection and analysis, decision to publish, or preparation of the manuscript.

**Competing interests:** The authors have declared that no competing interests exist.

*elegans* from its single-cell transcriptome, and show a negative correlation between interactions and intercellular distances, which is driven by a combination of ligand-receptor pairs following spatial patterns across the *C. elegans'* body, i.e., the spatial code. Thus, our framework uncovers molecular features crucial to defining spatial cell-cell interactions across a whole body; a strategy that can be readily applied in higher organisms.

## Introduction

Cell-cell interactions (CCIs) are fundamental to all facets of multicellular life. They shape cellular differentiation and the functions of tissues and organs, which ultimately influence organismal physiology and behavior. CCIs often take the form of secreted or surface proteins produced by a sender cell (ligands) interacting with their cognate surface proteins in a receiver cell (receptors).The nature of CCIs is constrained by the distance between interacting cells [1–3], and, in turn, CCIs follow spatial patterns of interaction [4]. These patterns are important since they allow CCIs to define cell location and community spatial structure [3,5]. For instance, some molecules mediating CCIs form gradients that serve as a spatial cue for other cells to migrate [6,7]. In addition, co-occurrence of ligands and receptors are strongly defined by their spatial neighborhoods [8], and cells can use these signals to sense spatial proximity to other cells [3]. Thus, it is reasonable to speculate that there is a spatial code embedded in ligand-receptor (LR) interactions across the body of multicellular organisms; a code that encodes spatial information and defines the distribution of cells in tissues and organs.

CCIs can be inferred from the gene expression levels of ligands and receptors [9]. Although spatial information is lost during tissue dissociation in conventional bulk and single-cell RNA-sequencing technologies (scRNA-seq) [10], inferring CCIs from transcriptomics can help elucidate how multicellular functions are coordinated by both the molecules mediating CCIs and their spatial context. Indeed, previous studies have proven that gene expression levels still encode spatial information that can be recovered by adding information such as protein-protein interactions and/or microscopy data [10–13]. For example, RNA-Magnet inferred cellular contacts in the bone marrow by considering the coexpression of adhesion molecules present on cell surfaces [12], while ProximID used gene expression coupled with microscopy of cells to construct a spatial map of cell-cell contacts in bone marrow [11]. Thus, we propose that CCIs inferred from transcriptomics could be extended to assess whether one can find, in RNA, a spatial code of intercellular messages that defines spatial organization and cellular functions across the whole body of a multicellular organism.

*Caenorhabditis elegans* is an excellent model for studying CCIs in a spatial context across a whole body [14]. This animal has fewer than 1,000 somatic cells stereotypically arranged across the body, whose locations have been described in a 3D atlas [15]. Despite the small number of cells, the intercellular organization in *C. elegans* shows complexity comparable to higher-order organisms. Taking advantage of these features, here we use scRNA-seq data from *C. elegans* to compute CCIs and assess which ligand-receptor pairs could govern an intercellular spatial code across the body. For this purpose, single-cell transcriptome data [16] were integrated with a 3D-atlas of cells of *C. elegans* [15], while we built the most comprehensive list of ligand-receptor interactions in *C. elegans* for CCI analyses. Next, we compared our CCI predictions to literature and found them consistent with previous studies independently reporting relevant roles of the identified LR interactions as encoders of spatial information. Additionally, we experimentally tested uncharacterized CCIs, and validated *in situ* that adjacent cells co-express the LR pairs computationally inferred to contribute to the spatial code. Thus, together, we

demonstrate that single-cell RNAseq data can be used to define a genotype-spatial phenotype link for the whole body in a multicellular organism.

## Results

### Computing cell-cell interactions

A first step to study cell-cell interactions can be to reveal active intercellular communication pathways from the coexpression of the corresponding LR pairs in any particular pair of cells. Communication scores can be assigned to each LR pair based on the RNA expression levels of their encoding genes in a given pair of sender and receiver cells [17–22]. Communication scores are then aggregated into an overall CCI score for each pair of cells, often represented by the number of active (expressed) LR pairs (LR Count score [18]), and in other cases by the sum of the LR expression product (ICELLNET score [23]). Higher numbers of active LR pairs and higher sum of expression LR levels can represent stronger cell-cell interactions [9]. However, these methods disregard that a high CCI score could result just by chance when one of the interacting cells promiscuously expresses many different ligands and/or receptors, or when the expression levels of a few LR pairs are too high, respectively. In contrast, we propose a novel CCI score that is based on the idea that high CCI scores should represent a high but also specific complementarity in the production of ligands and receptors between the interacting cells (Fig 1).

The specific complementarity captured by our Bray-Curtis score is also intended to represent a cell-cell potential of interaction that may respond to or drive intercellular proximity. Cells can sense the number of receptors that are occupied by signals from surrounding cells [3,24] and higher occupancy can indicate greater proximity of communicating cells [25]. Thus, our score is computed from the mRNA expression of ligands and receptors in pairs of interacting cells in a way that accounts for the usage fraction of the total number of expressed ligands and receptors (Fig 1A). The main assumption of our CCI score is that more proximal cells co-express more complementary ligands and receptors between the pair of cells. In other words, for any given pair of cells, cells are defined as closer when a greater fraction of the ligands produced by one cell interacts with cognate receptors on the other cell and vice versa, as this increases their potential of interaction in an undirected manner (Fig 1B).

To facilitate the implementation of our computational framework to predict a spatial code of CCIs and perform other general CCI analyses that do not rely on spatial information, we developed *cell2cell*. This open source tool infers intercellular interactions and communication using any gene expression matrix and list of LR pairs as inputs (https://github.com/earmingol/cell2cell), and depending on the purpose of a study, *cell2cell* also allows using other CCI scores beyond our Bray-Curtis score (e.g., LR Count and ICELLNET scores).

### Cell-type roles and spatial properties are captured by computed cell-cell interactions

To assess whether our Bray-Curtis score captures spatial properties associated with intercellular distances, we used *C. elegans* data since, among other relevant characteristics, this model organism has a stereotypical distribution of cells across its whole body that has been extensively studied through microscopy, and reported for 357 of its individual cells in a 3D atlas [15]. To compute the complementarity of interaction between *C. elegans* cells, an extensive list of functional LR interactions is needed. However, while much is known about *C. elegans*, knowledge of its LR interactions remains dispersed across literature or contained in protein-protein interaction (PPI) networks that include other categories of proteins. Thus, we first

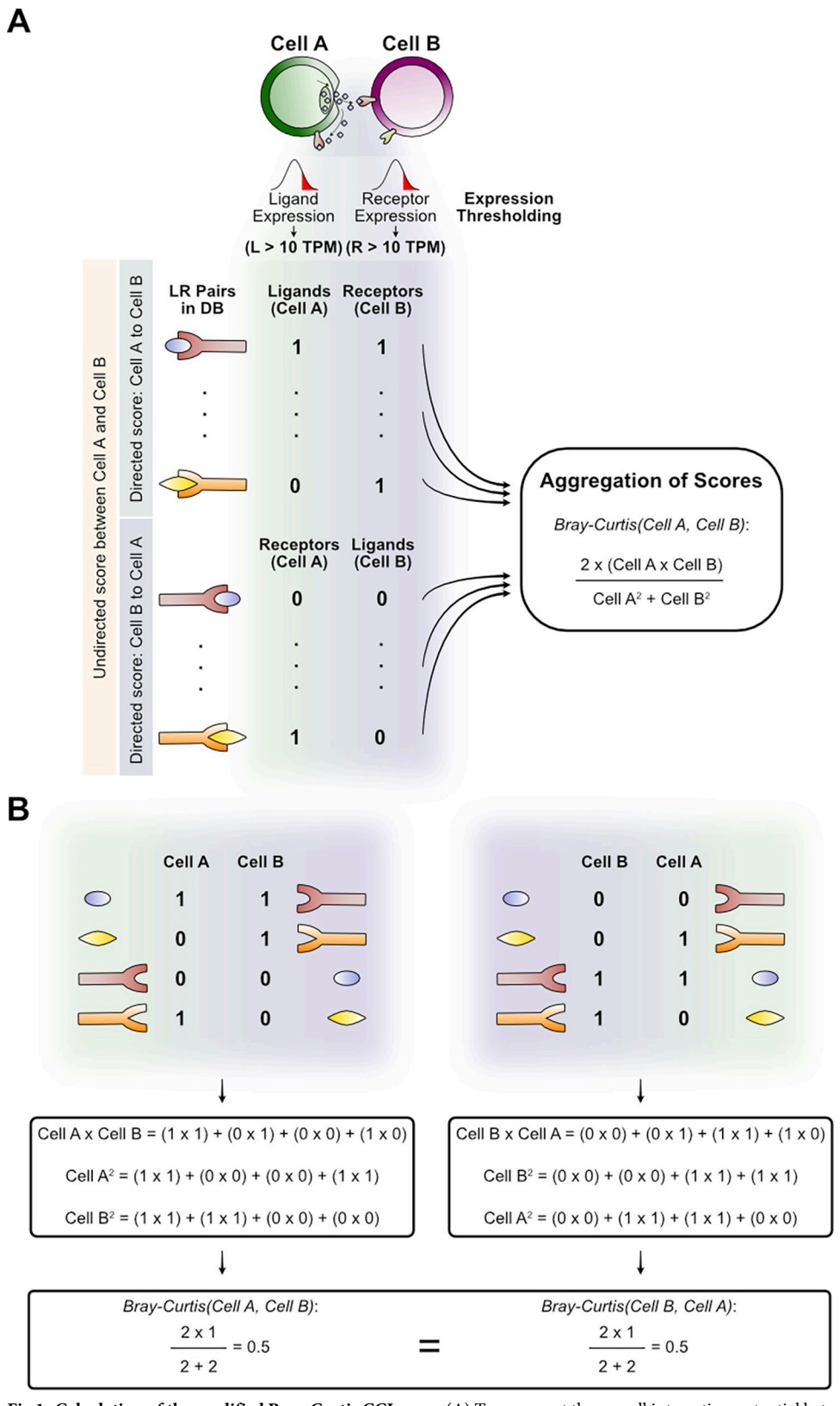

**Fig 1. Calculation of the modified Bray-Curtis CCI score.** (A) To represent the overall interaction potential between cell A and cell B, our CCI score is computed from two vectors representing the ligands and receptors independently

expressed in each cell. If only the ligands from one cell and the cognate receptors on the other are considered ("Cell A to Cell B" half or "Cell B to Cell A" half, independently), the score would be a directed score for representing the interaction (one cell is the sender and the other is the receiver). However, our score is undirected by considering both ligands and receptors of each cell to build the vector (both halves simultaneously, indicated with the yellow rectangle on the left). Thus, the vector of each cell is built with both directed halves of molecule production (e.g., top half possess ligands of cell A while the bottom half considers its receptors, generating a unique vector with both the ligands and the receptors of cell A). (B) Toy examples for computing our score for the interaction of Cell A and Cell B. Here, both possible directions of interaction are represented to show that they result in the same (undirected) score.

generated a list of 245 ligand-receptor interactions in *C. elegans* (S1 Table). Next, we used this list to determine the presence or absence of mRNAs encoding ligands and receptors in each cell identified in the single-cell transcriptome of *C. elegans* [16]. Briefly, this dataset takes a matrix of gene expression data with the aggregated values from all individual cells with the same annotation. Of the 27 cell types identified across the body of *C. elegans*, we considered only the 22 cell types that we were able to assign a spatial location in a previously published 3D atlas [15]. After integrating this aggregated single-cell transcriptomic data with the list of LR pairs, we inferred the active (expressed) LR pairs in all pairs of cell types by using a binary communication score (S2 Table). Next, we aggregated the respective communication scores for each cell pair with our Bray-Curtis metric, generating the first predicted network of CCIs in *C. elegans* that measures the complementarity of interacting cell types given their active LR pairs (Fig 2A).

After determining the potential for interaction between every pair of cell types from the single-cell transcriptome of *C. elegans*, we grouped the different cell types based on their interactions with other cells through an agglomerative hierarchical clustering (Fig 2A). This analysis generated clusters that seem to represent known roles of the defined cell types in their tissues. For instance, neurons have the largest potential for interactions with other cell types, especially with themselves and muscle cells. This suggests that these cell types use a higher fraction of all possible communication pathways, which is consistent with the high molecule interchange that occurs at the neuronal synapses and the neuromuscular junctions [26]. Also, seemingly in line with basement membranes surrounding germline cells and physically constraining their ability to communicate with other cell types [27,28], germline cells have the lowest CCI potential with other cell types. Thus, the results suggest that our method may be properly capturing the nature of the interactions between vastly different cell pairs.

We further observed that pairs of interacting cells tend to be grouped by the sender cells (i.e., those expressing the ligands), but not by the receiver cells (i.e., those expressing the receptors) (Figs 2B and S1). Remarkably, our result is consistent with previous findings that ligands are produced in a cell type-specific manner by human cells, but receptors are promiscuously produced [29]. While the study used a network-based clustering of ligands and receiver cell connections, we used UMAP [30,31] to visually summarize the Jaccard similarity [32] between pairs of interacting cell types, indicating this similar result from two different approaches could be biologically meaningful. Correspondingly, the coexpression of ligands and their cognate receptors follows a more similar behavior in cell pairs where the sender cells are of the same type, while the receiver cell types can be disregarded (S2 Fig).

Using the overall CCI scores computed for the cell-cell pairs in *C. elegans*, we next evaluated the ability of our Bray-Curtis score to separate distinct ranges of intercellular distances (short, mid, and long range, as defined in N1 Fig in S1 Text). To measure this ability, a classifier was trained by using the CCI scores as inputs and the intercellular-distance categories as outputs, and the performance was evaluated through a Receiver Operating Characteristic (ROC) curve and its area under the curve (AUC). In this regard, the Bray-Curtis score performed better

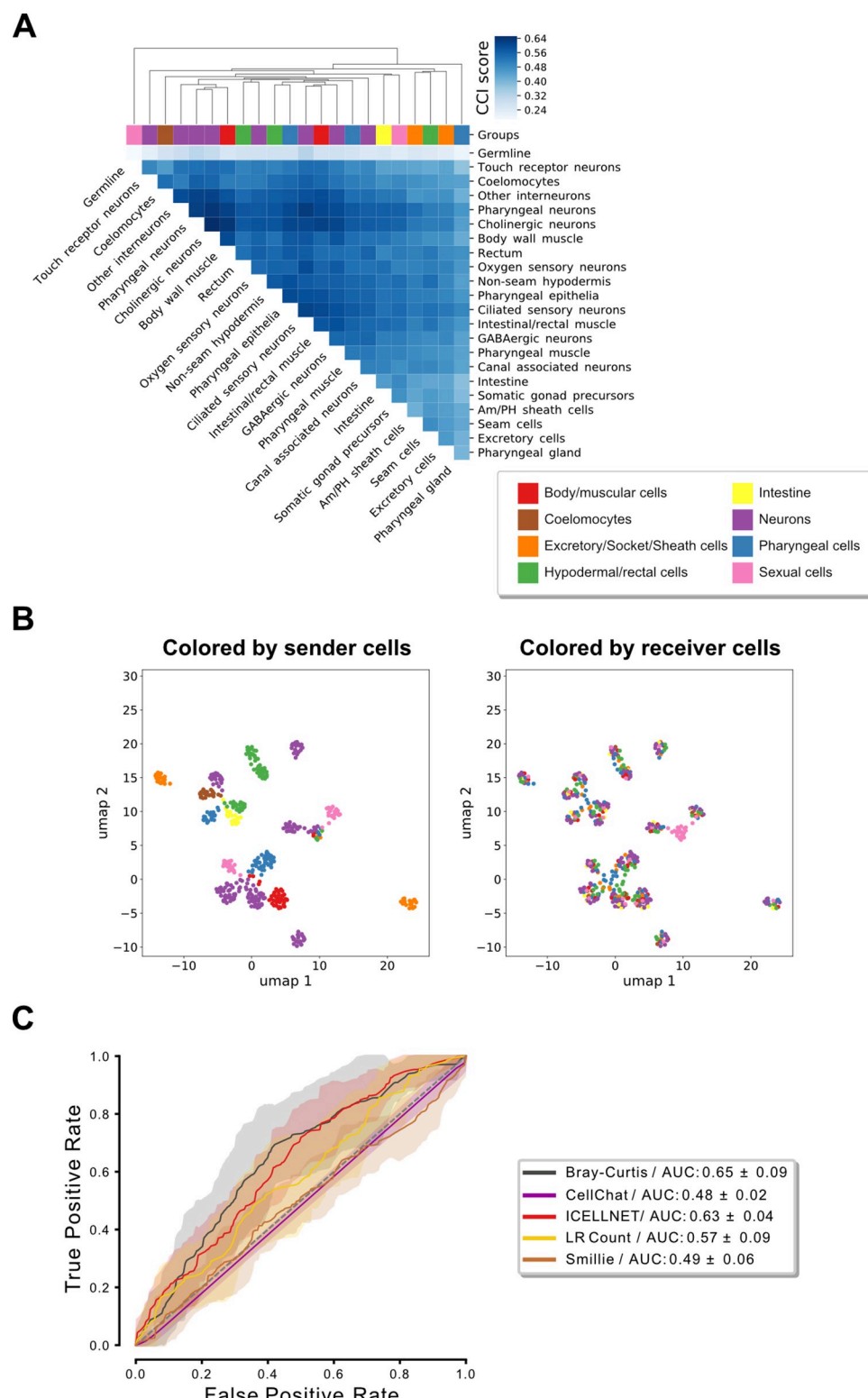

**Fig 2. Cell-cell interactions and communication in *C. elegans*.** (A) Heatmap of CCI scores obtained for each pair of cell types using the curated list of LR pairs. An agglomerative hierarchical clustering was performed on a dissimilarity-like metric by taking the complement (1-score) of CCI scores, disregarding autocrine interactions. Cell types are colored by their lineages as indicated in the legend. Lineages and colors were assigned previously [16]. (B) UMAP visualization of CCIs. Dots represent pairs of interacting cells and they were projected based on their Jaccard distances,

which were computed from the LR pairs expressed in the directed interactions between cells (one cell is producing the ligands and the other the receptors). Dots are colored by either the sender cell (left) or the receiver cell (right), depending on their lineages as indicated in the legend of (A). A readable version of the data used for this projection is available in S2 Table, where names of LR pairs and their communication scores are specified for each cell pair. Another UMAP visualization based on a more appropriate similarity metric is available in S1 Fig, which uses the Rand index that accounts for both active and inactive LR pairs. Using the Rand index still represents the same behavior of sender cells driving similarities. (C) Receiver operating characteristic (ROC) curves of random forest models for classifying cell-cell pairs from their CCI scores computed with different approaches as indicated in the legend. These models predict the intercellular distance range (short-, mid-, or long-range distance, as defined in the N1 Fig in S1 Text). For each classifier, the mean (solid line) ± standard deviation (transparent area) of the ROCs were computed with 3-fold stratified cross validations. The area under the curve (AUC) for the ROC curves is shown in the legend, detailing the mean ± standard deviation across all distance-range classifications. Separate evaluations for the distance ranges are provided in S3 Fig.

than a random model (avg. AUC of 0.65, Figs 2C and S3). In addition, we compared our score with other overall CCI scores, including those aggregated from binary communication scores, such as the number of active LR pairs (LR Counts) and the cell-type specific probability (Smillie) [33] and continuous-based scores, such as the sum of the LR expression product (ICELL-NET) [23] and the weight of significant LR pairs (CellChat) [34]. Under similar conditions of comparison, our Bray-Curtis score resulted to be the score that better separates intercellular-distance ranges, even slightly higher than the ICELLNET score (avg. AUC of 0.63), followed by the LR count–the most employed overall CCI score–(avg. AUC of 0.57). Interestingly, CCI scores based on permutations (CellChat and Smillie) had the lowest performance in separating intercellular distance ranges (avg. AUC ~0.5). However, the strength of permutation-based scores is that they better identify cell-type specific LR pairs and reduce the number of false positives in this regard, while they disregard LR pairs that are shared across multiple cell types. Thus, spatial proximity seems to be encoded by activation/inactivation of signaling mechanisms that are shared across multiple cell types rather than in very specific cell-type pairs.

## Signaling pathways involved in spatial patterning underlie the anticorrelation between cell distance and interaction potential

After validating that our score distinguishes spatial properties of cell-cell interactions, we further assessed the assumption that larger physical distances would decrease the potential of cells to interact. Thus, we evaluated the relationship between our undirected CCI score and the Euclidean distance between cell pairs (S1 Text). As expected, the correlation coefficient was negative (Spearman = -0.21; P-value = 0.0016). However, although negative, the anticorrelation is weak. Therefore, we hypothesized that there is a subset of key LR pairs encoding spatial organization. To identify this LR subset, we used a genetic algorithm (GA), that is based on natural evolution, to select a subset of our initial list of LR pairs that maximizes the anticorrelation between the CCI scores and the Euclidean distances (S1 Text). Using this approach, we found 100 candidate subsets from independent runs that led to an average Spearman coefficient of -0.67 ± 0.01. The genetic algorithm-optimized subsets of LR pairs (hereinafter referred to as 'initial GA-LR pairs') may therefore constitute good predictors of biological functions driving or sustaining intercellular proximity.

Using our literature-based functional annotations (see column "LR Function" in S1 Table), we next investigated the specific biological roles of the initial GA-LR pairs. Specifically, we computed the relative abundance of each functional annotation within each initial GA-LR pair (S4 Fig). Considering the relative abundances in our complete list containing the 245 LR pairs as the expected values, we performed a two-tailed Wilcoxon signed-rank test to evaluate whether the relative abundance of each signaling function either increased or decreased across

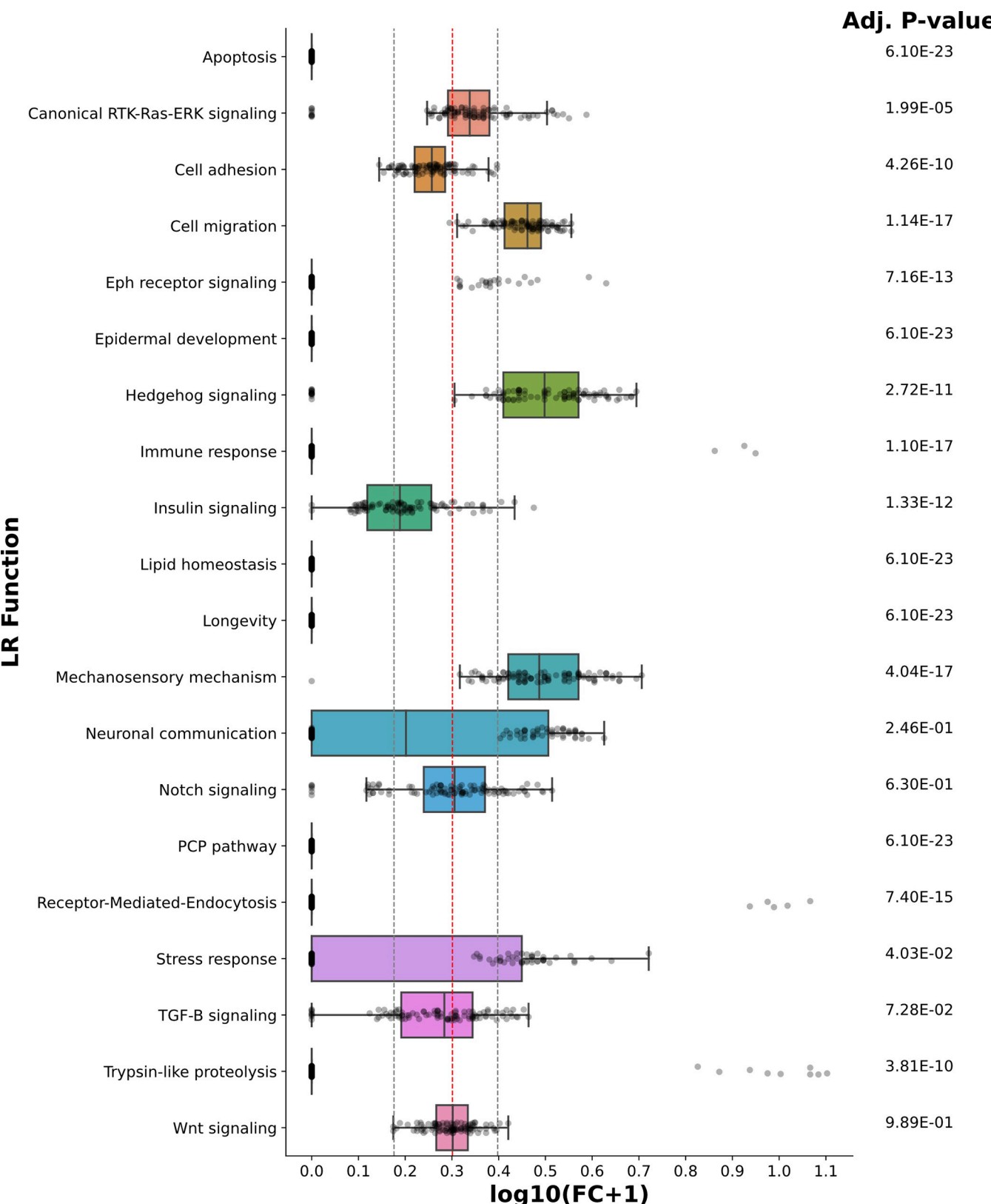

**Fig 3. Changes in the relative abundances of signaling functions across initial GA-LR pairs.** Boxplots summarizing the changes of the relative abundances for each of the signaling functions that LR pairs are associated with (y-axis). Changes were computed from the fold change (FC) between the relative abundance

in each of the 100 runs of the genetic algorithm (GA) with respect to the corresponding relative abundance in the complete list of LR pairs (S1 Table), and shown as the log10(FC+1) transformation (x-axis). Here, relative abundance is the number of LR pairs involved in a given pathway with respect to the total number of LR pairs in the list of GA-LR pairs. A two-tailed Wilcoxon's test was performed to evaluate the significance of the changes. An adjusted P-value is reported to the right of each boxplot (FDR < 1%). All GA runs are shown in each boxplot (gray dots); dashed-gray lines indicate a change of at least 50% either decreasing (left line, FC = 0.5) or increasing (right line, FC = 1.5), while the dashed-red line indicates the value of no change (FC = 1).

all GA runs (Fig 3). Remarkably, LR pairs involved in cell migration, Hedgehog signaling, mechanosensory mechanisms, and canonical RTK-Ras-ERK signaling increased their relative abundance in the resulting subsets from the GA runs. Thus, the GA prioritizes LR pairs associated with processes such as cell patterning, morphogenesis, and tissue maintenance [35].

Considering that the GA is a non-deterministic approach, different optimal solutions can be obtained from independent runs. Thus, we next looked for a consensus set of LR pairs among all optimal solutions generated by our GA (S1 Text). This resulted in a list of 37 LR pairs (S3 Table), hereinafter referred to as GA-LR pairs, yielding a Spearman coefficient of -0.63 (P-value = 2.629 x $10^{-27}$) between the CCI scores and Euclidean distances; a correlation that is not a result of randomly selecting LR pairs (P-value = 0.0002, permutation tests detailed in S1 Text). While the CCI scores computed from the complete list of LR pairs led to functional interactions of cell types (Fig 2A), the GA-LR pairs seem to group cell types by more specific associations that may be attributable to their spatial localization (Fig 4A). For example, the complete list grouped neurons and muscles together, while the GA-LR pairs increased the specificity of this association by grouping both excitatory and inhibitory neurons (cholinergic and GABAergic neurons, respectively) directly with all muscles. Furthermore, these GA-LR pairs group all cells composing the pharynx (pharyngeal gland, epithelia, muscle and neurons) together. Another interesting observation was the high CCI score between oxygen sensing neurons and intestinal cells, consistent with the extensive communication between these cells to link oxygen availability with nutrient status [36–38]. Thus, the LR interactions prioritized by the GA capture cellular properties that define not only intercellular proximity, but more importantly, cell-community structure of tissues and organs. Moreover, some participating ligands and receptors are expressed in few cell types while others are found in most cell types (Fig 4B), suggesting that our algorithm captures both communication between specific pairs of cells and more promiscuous interactions. Thus, we hypothesized these PPIs represent a spatial code that can encode different spatial proximities.

## The GA-LR pairs define a spatial code of intercellular interactions along the body

To start assessing the hypothesis that the GA-LR pairs represent a spatial code, we performed an enrichment analysis of CCI mediators along the body of *C. elegans*. We first divided the *C. elegans* body in 3 sections, encompassing different cell types (Fig 5A). We then computed all pairwise CCIs within each section and counted the number of times that each LR pair was used. With this number, we performed a Fisher's exact test on each bin for a given LR interaction. We observed enrichment or depletion of specific LR pairs in different parts of the body (Fig 5B). Interestingly, we observed LR pairs enriched only in one section and depleted in the others and vice versa (Table 1), following a pattern mostly congruent with existing experimental data (S4 Table). For instance, *col-99* shows prominent expression in the head, especially during L1-L2 larvae stages of development [39], while LIN-44 is secreted by hypodermal cells exclusively in the tail during larval development [40,41], both cases coinciding with the results in Table 1. Although a few spurious results emerged, they are mainly associated with the limitations of the current scRNAseq methods and their analysis tools (see Discussion section).

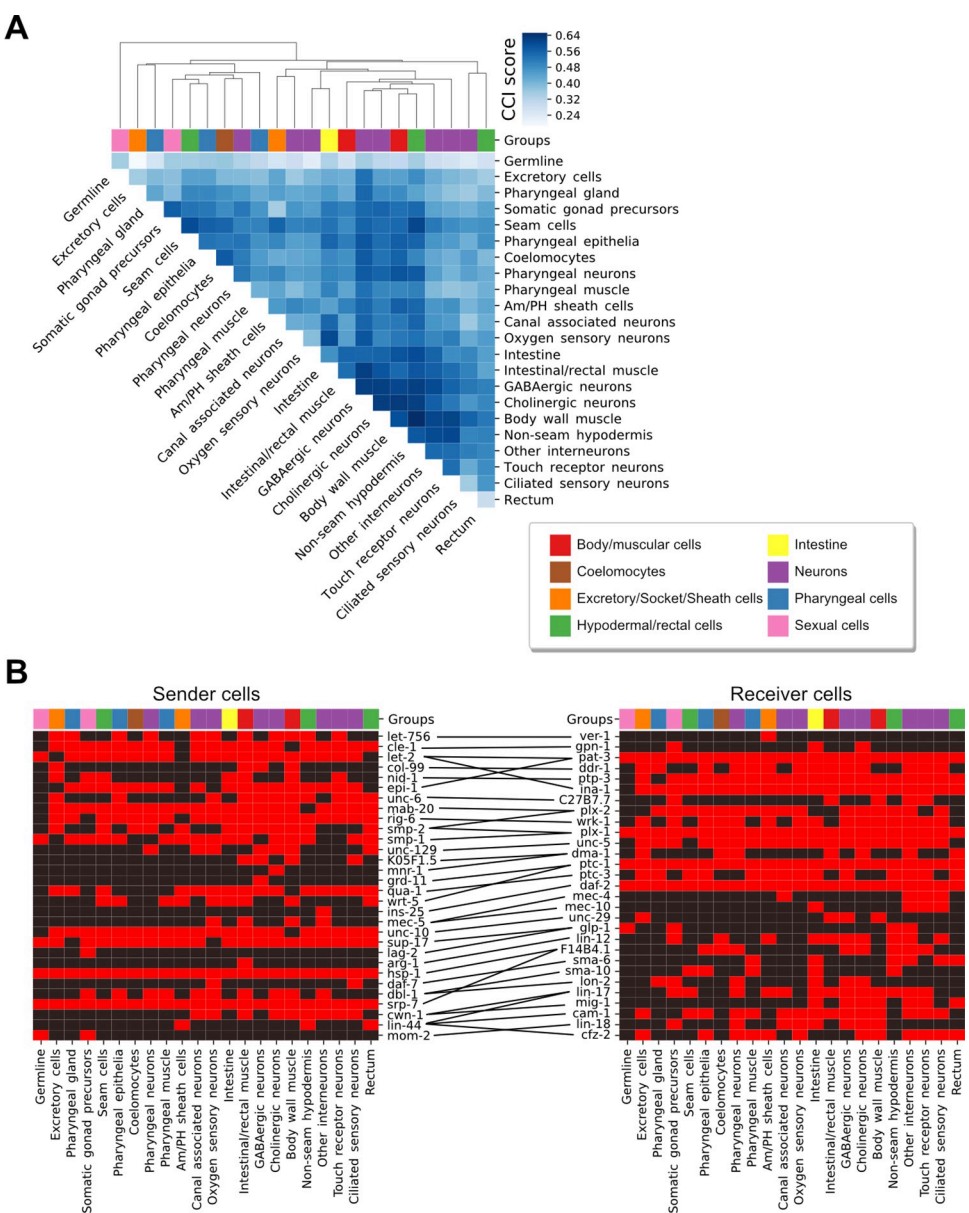

**Fig 4. CCI analyses based on LR pairs associated with intercellular distances.** (A) Heatmap of CCI scores obtained for each pair of cells using the consensus GA-LR pairs. An agglomerative hierarchical clustering was performed on a dissimilarity-like metric by taking the complement of CCI scores (1-score), excluding autocrine interactions. Cell types are colored by their lineage groups as indicated. (B) Heatmaps representing the presence or absence of ligands (left) and receptors (right) after expression thresholding (>10 TPM) in sender and receiver cells, respectively. Lines at the center connect ligands with their cognate receptors according to the GA-selected interactions. Cell types are colored as in (A).

Therefore, the *col-99* and *lin-44* examples support the notion that our strategy captures the spatial distribution of gene expression and therefore of CCIs across the *C. elegans* body.

To better understand the importance of the GA-LR pairs in identifying spatially-constrained CCIs, we searched for LR pairs enriched or depleted across all cell pair interactions in any of the different distance-ranges of communication. We found five LR pairs that were either enriched or depleted in at least one of the three distance ranges given the corresponding

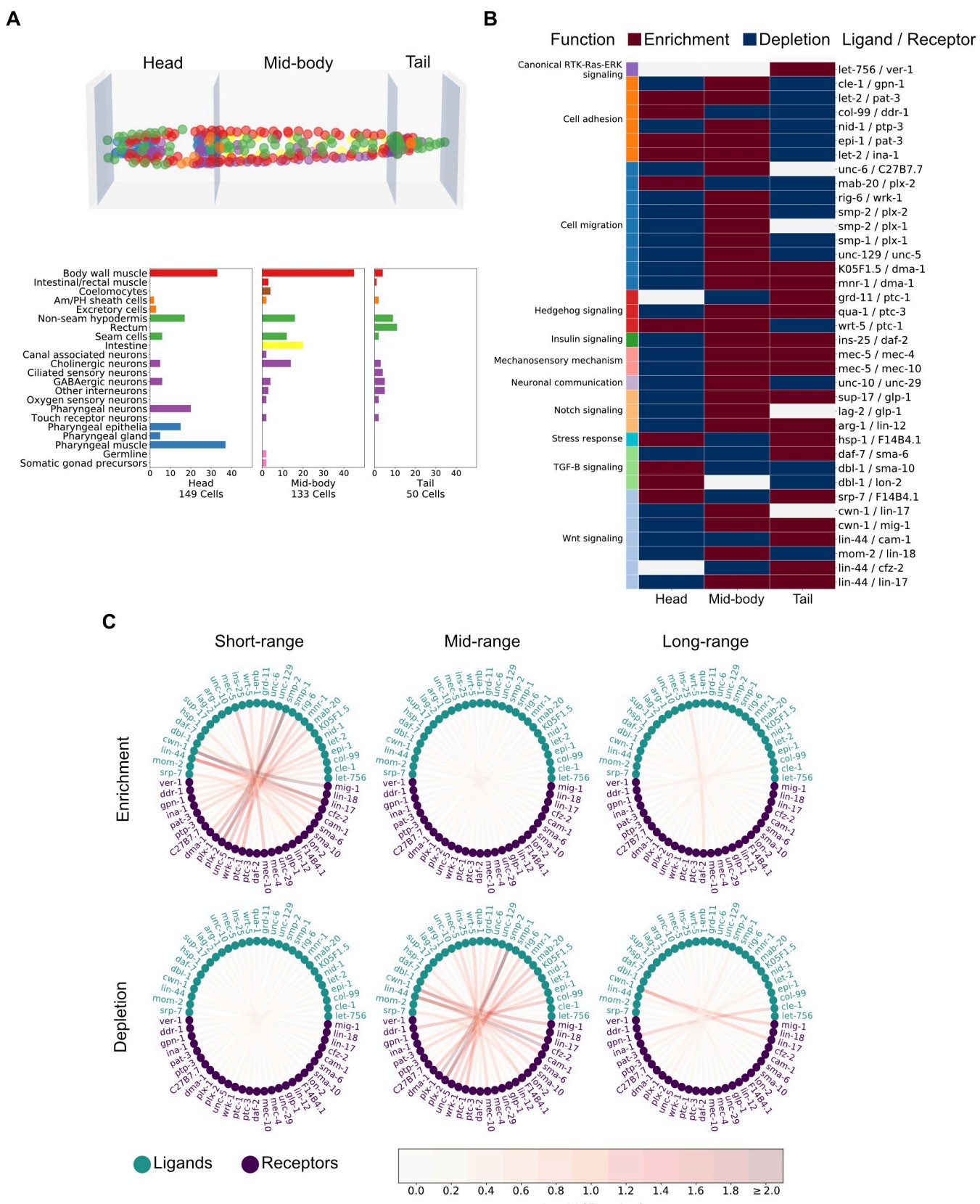

**Fig 5. Spatial enrichment and depletion of communication pathways.** (A) To study the anteroposterior use of communication pathways, the body of *C. elegans* was divided into three sections along the anteroposterior axis (top) and cell-type composition of each section (bottom) given a previously published 3D atlas. The mid-body section is defined by the presence of the intestine cells, and the head and tail are the anterior and posterior sections to it, respectively. Cells in the 3D atlas (top) are colored according to the cell types as delineated in barplots (y-axis, bottom). (B) Enrichment/depletion (FDR < 1%) of ligand-receptor pairs (y-axis) in each of the three sections (x-axis), calculated from their usage across all pairs of cells of each section. Communication pathways are also colored by their annotated functions (left column). (C) Circos plots for representing the importance of cell-cell communication occurring at different ranges of distance. A Fisher exact test was performed to find enriched/depleted LR pairs among all pairs of cells for a given proximity. The ranges of distances were defined as explained in Fig N1C. Nodes represent ligands or receptors and edges connect those ligands and receptors that interact in the GA-LR pairs (S3 Table). The color of the nodes represent whether they are ligands or receptors and the color of the edges indicate the negative value of the logarithmic transformation on the Benjamini-Hochberg adjusted P-values, according to the colored bar at the bottom. Interactions that resulted significantly enriched or depleted (FDR < 1%) are equivalent to the color assigned to a value of 2.0 or bigger.

pairs of cell types (FDR < 1%) (Fig 5C). Three of these LR pairs are associated with Wnt signaling (*lin-44*/*cfz-2*, *cwn-1*/*lin-17* and *cwn-1*/*mig-1*) and the other two with cell migration (*smp-2*/*plx-1* and *smp-2*/*plx-2*). Members of the Wnt signaling act as a source of positional information for cells [3]. For example, in *C. elegans*, *cwn-1* and *lin-44* follow a gradient along its body, enabling cell migration [42–45]. While semaphorins (encoded by *smp-1*, *smp-2* and *mab-20*) and their receptors (plexins, encoded by *plx-1* and *plx-2*) can control cell-cell contact formation [46], and their mutants show cell positioning defects, especially along the anterior/posterior axis of *C. elegans* [47,48], affecting axon guidance, cell migration [49], epidermal and vulval morphogenesis [50,51]. Thus, the GA-LR pairs may influence local or longer-range interactions and help encode intercellular proximity.

The spatial code can be considered as the biochemical signals used by cells to build a physical network of interactions. As such, another natural question is whether groups of signals in the GA-LR pairs are enriched in the distinct distance ranges of interactions. By annotating every ligand-receptor pair with a location type where the ligand act (ECM-component, membrane-bound, or secreted), we also assessed if any of these kinds of LR pairs are more likely to participate in the intercellular interaction given the distance range of the cells. We observe that

**Table 1. Ligand-receptor interactions enriched or depleted in one body section and depleted or enriched in the rest.**

| Interactions enriched in a body section and depleted in the rest | | | Interactions depleted in a body section and enriched in the rest | | |
|---|---|---|---|---|---|
| Ligand | Receptor | Section | Ligand | Receptor | Section |
| col-99 | ddr-1 | Head | K05F1.5 | dma-1 | Head |
| mab-20 | plx-2 | Head | mnr-1 | dma-1 | Head |
| dbl-1 | sma-10 | Head | qua-1 | ptc-3 | Head |
| cle-1 | gpn-1 | Mid-Body | ins-25 | daf-2 | Head |
| nid-1 | ptp-3 | Mid-Body | mec-5 | mec-4 | Head |
| rig-6 | wrk-1 | Mid-Body | mec-5 | mec-10 | Head |
| smp-2 | plx-2 | Mid-Body | sup-17 | glp-1 | Head |
| smp-1 | plx-1 | Mid-Body | arg-1 | lin-12 | Head |
| unc-129 | unc-5 | Mid-Body | cwn-1 | mig-1 | Head |
| unc-10 | unc-29 | Mid-Body | lin-44 | lin-17 | Head |
| mom-2 | lin-18 | Mid-Body | hsp-1 | F14B4.1 | Mid-Body |
| daf-7 | sma-6 | Tail* | srp-7 | F14B4.1 | Mid-Body |
| lin-44 | cam-1 | Tail | let-2 | pat-3 | Tail |
| | | | epi-1 | pat-3 | Tail |
| | | | let-2 | ina-1 | Tail |
| | | | wrt-5 | ptc-1 | Tail |

* See the discussion section for details about this prediction.

ECM-component LR pairs are more likely to be used than the other types in the mid-range interactions (odds ratio = 1.23, P-value = 0.0135), while they are less likely to be used than other types in short-range interactions (odds ratio = 0.84, P-value = 0.0229). In contrast, secreted LR pairs were slightly overrepresented with respect to the other location types in short-range interactions (odds ratio = 1.19, P-value = 0.0149) and underrepresented in mid-range interactions (odds ratio = 0.82, P-value = 0.0093). While membrane-bound LR pairs did not show any over- or underrepresentation, we noticed that cases such as *grd-11/ptc-1*, *lag-2/glp-1*, *arg-1/lin-12*, and *mnr-1/dma-1* are used by few cell-cell pairs, and mainly participate in short-range interactions (S5 Fig). Membrane-bound interactions may involve more general mechanisms of cells and be passively acting, meaning that their co-occurrence may respond to the proximity influenced by ECM-component and secreted LR pairs. Thus, the spatial code seems to be partially driven by the nature of the LR pairs, encoding biologically meaningful information behind the correlation between our Bray-Curtis score and intercellular distance.

Our hypothesis that key LR pairs encode spatial CCI information also implies the assumption that cell-type localization is crucial for organismal phenotypes and functions. Thus, we performed a phenotype enrichment analysis for *C. elegans* [52] using the GA-LR genes (sampled genes) and the complete LR pair list (background genes). The 'organ system phenotype' was the only enriched term, with odds ratio of 4.13 (Fisher's exact test; adj. P-value = 0.0029). According to WormBase [53], this term represents a generalization for phenotypes affecting the morphology of organs, consistent with the clustering of cell types by their tissue lineage groups when considering genes associated with this phenotype (S6 Fig). Thus, our GA-LR pairs seem to encode more general relationships, including an association between CCIs and organ organization across the body, which in higher organisms could have an impact leading to diseases when perturbed.

## GA-LR pairs are proximally expressed in *C. elegans*

So far, our computational framework seems to be able to identify LR interactions driving the spatial organization of cell-cell interactions. Therefore, based on the precedents presented here, especially the strong anticorrelation between our CCI score and the intercellular distance, we expected that the ligand and the receptor in some of the GA-LR pairs to be expressed in proximal cells. To test whether the LR pairs selected by our algorithm are actually co-expressed in proximal cell pairs, we searched the literature for established interactions in addition to experimentally testing new CCIs. We found several LR pairs with known expression patterns in *C. elegans* that coincide with the predictions of our algorithm (reported elsewhere; and summarized in S4 Table). Furthermore, we used single-molecule Fluorescent *In Situ* Hybridization (smFISH) to test whether previously uncharacterized LR pairs are co-expressed in adjacent cells as predicted by our model. Specifically, we focused on the uncharacterized interactions between *arg-1/lin-12*, *let-756/ver-1* and *lin-44/lin-17* (see Methods for selection criteria).

Confirming the predictions of our algorithm, we found the ligand and receptor genes expressed in spatially proximal cells. In particular, our *in situ* results confirmed our computational prediction that *arg-1* and *lin-12* are proximally expressed in the intestinal/rectal muscle and the non-seam hypodermal cells (Fig 6A). We also confirmed that *let-756* is expressed in the non-seam hypodermal cells of the head, proximally to *ver-1* in the amphid sheath cells (Fig 6B). Finally, *lin-44* in the non-seam hypodermal cells of the tail is proximally expressed to *lin-17* in the seam cells of the tail (Fig 6C). Additionally, we performed 3D projections of the smFISH images (S1–S3 Movie), which more clearly show the extent of the spatial adjacency of the cells expressing the cognate LR pairs. We also noticed that while *arg-1/lin-12* (S1 Movie) and *let-756/ver-1* (S2 Movie) were expressed in cells that were juxtaposed, *lin-44/lin-17* were

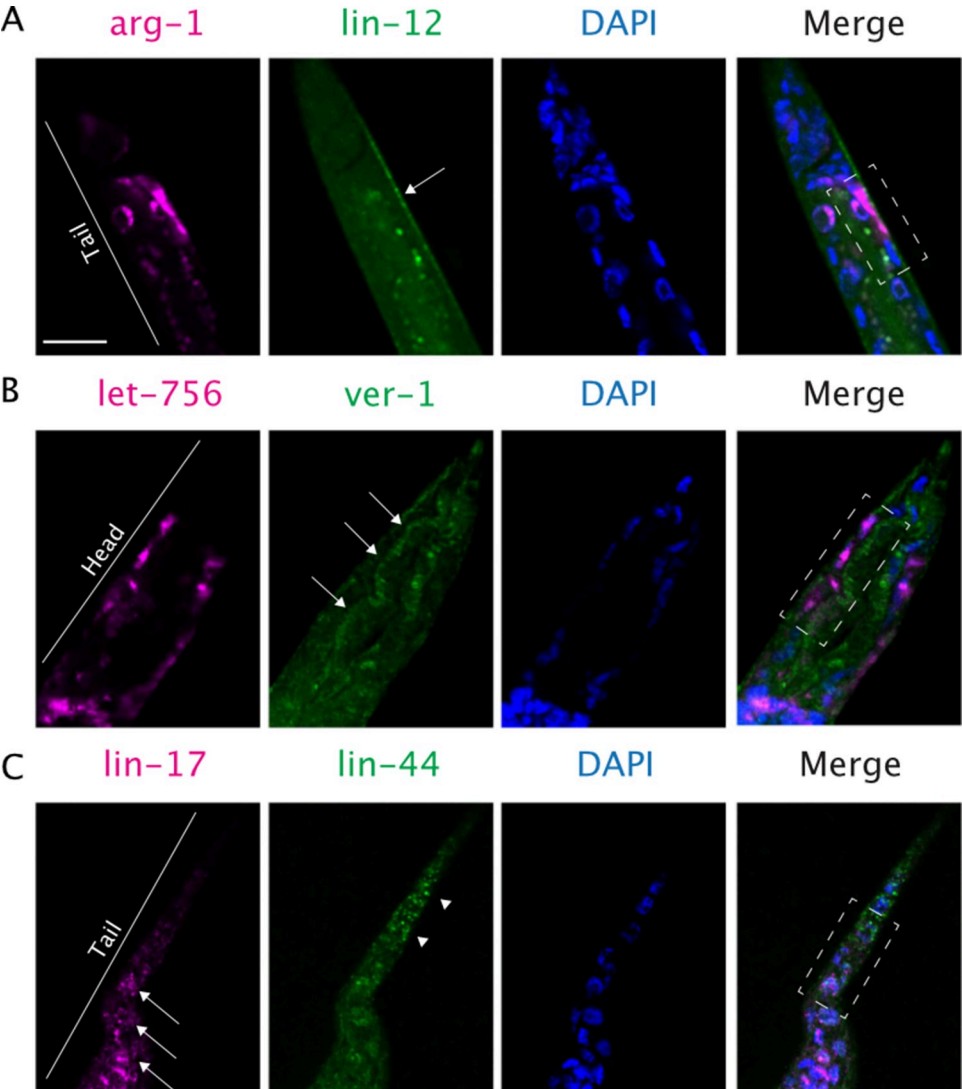

**Fig 6. Validation of the spatial expression of specific GA-LR pairs.** Single-molecule Fluorescent In Situ Hybridization of genes encoding three GA-LR pairs in *C. elegans* L2 larvae. (A) Intestinal/rectal cells expressing *arg-1* (magenta) and non-seam hypodermal cells (arrow) expressing *lin-12* (green) are adjacent (see rectangle in the merge channel and S1 Movie). (B) Non-seam hypodermal cells expressing *let-756* (magenta) and amphid sheath cells (arrows) expressing *ver-1* (green). Amphid sheath cells are surrounded by hypodermal cells (see rectangle in the merge channel and S2 Movie). (C) Seam cells in the tail (arrows) expressing *lin-17* (magenta) and non-seam hypodermal cells in the tail (arrowheads) expressing *lin-44* (green). The two genes are expressed in proximal cells (see ends of rectangle in the merge channel and S3 Movie). In all cases a DAPI staining was performed to distinguish cell nuclei. Scale bar = 10μm.

expressed in proximal but not necessarily juxtaposed cells (S3 Movie). Interestingly, LIN-44 is a secreted ligand [54] that was inferred to participate in multiple cell-cell pairs encompassing all distance-range interactions (S5 Fig). Therefore, the experimental observations are consistent with our inferences for the respective pairs of cells, which presented CCI scores among the highest values, being of 0.61, 0.58 and 0.64, respectively (Fig 4A). Thus, the results not only further support the notion that higher Bray-Curtis scores represent a higher potential of cells to be spatially proximal, but they also show that applying our computational framework is a hypothesis generator of unknown biology.

## Discussion

Here we present a computational strategy to quantify the potential of cells to interact and communicate, which we named *cell2cell*. Using scRNA-seq data and a list of validated and predicted LR pairs, this tool uses the gene expression level of the ligands and the receptors in all cells in the dataset to infer how they communicate. Furthermore, we implemented in *cell2cell* a new scoring function to compute an overall potential of cells to interact, the Bray-Curtis CCI score, which is intended to represent both interaction strength and intercellular proximity. By applying this approach to infer interactions in *C. elegans* we identified that this score can distinguish ranges of intercellular distances, and we further showed a negative correlation with intercellular distance. Thus, our computational framework is useful for associating phenotypes with CCIs in a space-dependent fashion.

By using a genetic algorithm to search for a combination of fewer LR pairs that could encode spatial information, we found a consensus subset of 37 LR pairs that enhanced the negative correlation of our Bray-Curtis score and intercellular distances. Specifically, it decreased the Spearman coefficient from -0.21 to -0.63. Importantly, this further indicates that specific LR pairs can encode spatial proximity of cells in *C. elegans*, supporting the notion that the GA-LR pairs impact organismal-level phenotypes. Furthermore, we compared our Bray-Curtis score–which is based on binary expression of ligands and receptors–to other CCI scores, including both binary- and continuous-expression based scores. Overall, our method performed better than the other scores (Fig 2C), but comparable to ICELLNET, a continuous-expression-based score. In this regard, continuous levels of LR-pair activation could provide further details missed with binary-based scores, but introducing biases associated with few LR pairs having high expression levels. We further ran the GA on the top-3 performer CCI scores (Bray-Curtis, LR Count, and ICELLNET), resulting in similar distributions of correlation across all independent GA runs (S7A Fig). In addition, the consensus LR lists in all cases resulted to be biologically comparable, presenting an important overlap (S7B Fig). This suggests that the GA approach captures biologically meaningful processes regardless of the scoring approach, and strengthens the evidence that a spatial code across the whole body is dependent on specific LR interactions.

To apply *cell2cell* in *C. elegans*, we needed a database of known LR interactions in this organism, which had not yet been reported. Hence, we collected LR interactions of *C. elegans* from the literature and databases of PPIs to build the most comprehensive database of LR interactions in *C. elegans* (S1 Table). We anticipate this will be a valuable resource and hypothesis generator for the study of CCIs in *C. elegans*, either at a spatial or functional level. In this regard, our CCI analysis based on this database identified a core set of LR pairs associated with spatial patterning in *C. elegans*. For example, we found that interacting cells were grouped based on cell type-specific production of ligands (Fig 2B), which is consistent with the principles underlying a communication network reported for human haematopoietic cells [29]. Our results are also consistent with previous experimental studies of *C. elegans* (S4 Table). For instance, the GA-driven selection of LR pairs prioritized mediators with a role in cell migration, Hedgehog signaling, mechanosensory mechanisms and canonical RTK-Ras-ERK signaling (Fig 3B). These GA-LR pairs also included LR interactions that are crucial for the larval development of *C. elegans*, especially of processes driven by Notch and TGF-β signaling, as well as cellular positioning, and organ morphogenesis, which are particularly active at the cognate stages of the datasets we used [55–58]. Thus, the GA-selected LR pairs are enriched in processes that contribute to defining the spatial properties of tissues and organs. Furthermore, some GA-LR pairs more likely act in short distance interactions (Fig 5C) and in specific body regions (Fig 5B), which may also be associated with the biochemical nature of these LR pairs

(S5 Fig). Therefore, the genetic algorithm prioritized a core list of LR pairs whose active/inactive combination seems to define a cellular spatial code across the *C. elegans* body.

Importantly, the LR pairs selected by the GA can affect different phenotypes that are related among them in *C. elegans*, suggesting that new biology can be inferred when including less-studied LR pairs. For instance, the GA-LR pairs enriched in short distance interactions (Fig 5C) include: 1) The LR pair composed of *smp-2*/*plx-1*, which mediates epidermal morphogenesis, as demonstrated by the defects in epidermal functions exhibited by *C. elegans* lacking *smp-2* [47]; and 2) *cwn-1*/*mig-1*, which mediates cell positioning, as demonstrated by the abnormal migration of hermaphrodite specific motor neurons in the mutants [44,59]. Additionally, by using smFISH we experimentally showed that mediators used by the cell pairs with high CCI scores (Fig 4A), such as *arg-1*/*lin-12*, *let-756*/*ver-1* and *lin-44*/*lin-12*, are expressed in spatially adjacent or proximal cells (Fig 6). While previous studies reported that hypodermal cells form a gradient of LIN-44 in the tail [60], and that LIN-44 can affect seam cell polarity through LIN-17 [61], the spatial proximity necessary for this LR pair to mediate a CCI had not been shown before. Although smFISH is not a direct proof of CCIs, mRNA co-expression serves as a good proxy for experimentally supporting those CCIs [9,62]. *lin-44*/*lin-17* is proximally co-expressed in tail hypodermal and seam cells (Fig 6C and S3 Movie). Thus, the previous reports and the smFISH results show congruence with the predictions of our algorithm. This not only increases the confidence in our approach and results, but it also exposes the potential of our computational framework to uncover LR interactions that were not previously studied in specific cell types.

Overall, our strategy captures mechanisms underlying the spatial and functional organization of cells in a manner that is consistent with prior and new experimental evidence (S4 Table and Fig 6). Nevertheless, our approach has some limitations. Conventional scRNA-seq does not preserve spatial information, so labeling cells in a 3D atlas by using cell types as annotated in a transcriptomic dataset might be a confounder. For example, *C. elegans* possesses sub-types of non-seam hypodermal cells, and their gene expression varies depending on their antero/posterior location. However, the scRNA-seq data set employed here pooled all non-seam hypodermal cell subtypes as one cell type, artificially generating a generic hypodermal seam cell with a uniform gene expression profile across the body. An illustrative case where this impacted our predictions is the expression of *lin-44*, which is exclusively expressed in hypodermal cells of the tail (Fig 6C) [42,45], but our method inferred that *lin-44* was also important in the mid-body (Fig 5B, pair *lin-44*/*lin-17*). Similarly, *daf-7* is expressed only in sensory neurons in the head [63]; however, our results show an enrichment in the tail (Table 1). This discrepancy is likely due to pooling the transcriptome of the two types of sensory neurons that express *daf-7*. Similarly, *ver-1* is expressed by amphid and phasmid sheath cells, which are located in the head and tail, respectively; however, these cells are annotated as the same cell type in the transcriptome: amphid/phasmid (Am/PH) sheath. Thus, the labeling of both groups of cells as Am/PH sheath cells could explain an enrichment of the *let-756*/*ver-1* interaction only in the tail (Fig 5B) even though it is an important communication also happening in the head (Fig 6). Therefore, relying only on conventional scRNA-seq enables us to infer the LR interactions that a pair of cells can theoretically use but may not actually use. These limitations of scRNA-seq may also explain the strong but imperfect correlation obtained between CCI scores and intercellular distances, which may be evaluated by spatial transcriptomics in future studies of CCIs [64,65]. In this regard, our approach can readily use this kind of technology to understand spatial properties of *C. elegans* or other organisms, even at the level of tissues instead of the whole body.

In summary, our computational framework combines the use of *cell2cell* and a GA to find a combination of LR pairs mediating overall CCIs that best correlates with the intercellular distances. As shown in this work, when considering spatial information, our approach is capable

of recovering spatial properties lost in the traditional transcriptomics methods, either bulk or single cell, which is important since these technologies are easier to access than the technologies preserving spatial properties (e.g., spatial transcriptomics). Also, as long as a pertinent objective function can be defined for the GA, our strategy can be used to identify LR pairs associated with phenotypes of interest. Thus, our strategy provides a framework for unraveling the molecular and spatial features of cell-cell interactions and communication across a whole animal body, and potentially their phenotypic consequences. Finally, while our approach can be extended to study the role of CCIs in physiological and diseased states in higher organisms, it is important to consider that faster algorithmic approaches than the GA could be applied for searching a spatial code in larger datasets, but with other computational limitations.

## Methods

### Single-cell RNA-seq data

A previously published transcriptome of 27 cell types of *C. elegans* in the larval L2 stage was used [16]. The cell types in this dataset belong to different kinds of neurons, sexual cells, muscles and organs such as the pharynx and intestine. We used the published preprocessed gene expression matrix for cell-types provided previously [16], wherein the values are transcripts per million (TPM).

### Intercellular distances of cell types

A 3D digital atlas of cells in *C. elegans* in the larval L1 stage, encompassing the location of 357 nuclei, was used for spatial analyses of the respective cell types [15]. Each of the nuclei in this atlas was assigned a label according to the cell types present in the transcriptomics dataset, which resulted in a total of 322 nuclei with a label and therefore a transcriptome. To compute the Euclidean distance between a pair of cell types, all nuclei of each cell type were used to compute the distance between all element pairs (one in each cell type). Then, the minimal distance among all pairs is used as the distance between the two cell types (N1A Fig in S1 Text). In this step, it is important to consider that this map is for the L1 stage, while the transcriptome is for the L2 stage. However, we should not expect major differences in the reference location of cells between both stages.

### Generating a list of ligand-receptor interaction pairs

To build the list of ligand-receptor pairs of *C. elegans*, a previously published database of 2,422 human pairs [18] was used as reference for looking for respective orthologs in *C. elegans*. The search for orthologs was done using OrthoDB [66], OrthoList [67] and gProfiler [68]. Then, a network of protein-protein interactions for *C. elegans* was obtained from RSPGM [69] and high-confidence interactions in STRING-db (confidence score > 700 and supported at least by one experimental evidence) [70]. Ligand-receptor pairs were selected if a protein of each interaction was in the list of ortholog ligands and the other was in the list of ortholog receptors. Additionally, ligands and receptors mentioned in the literature were also considered (S5 Table). Finally, a manual curation as well as a functional annotation according to previous studies were performed, leading to our final list of 245 annotated ligand-receptor interactions, encompassing 127 ligands and 66 receptors (S1 Table).

### Communication and CCI scores

To detect active communication pathways and to compute CCI scores between cell pairs, first it was necessary to infer the presence or absence of each ligand and receptor. To do so, we

used an expression threshold over 10 TPM as previously described [18]. Thus, those ligands and receptors that passed this filter were considered as expressed (a binary value of one was assigned). Then, a communication score of one was assigned to each ligand-receptor pair with both partners expressed; otherwise a communication score of 0 was assigned. To compute the CCI scores, a vector for each cell in a pair of cells was generated using their communication scores as indicated in Fig 1. These vectors containing the scores were aggregated into a Bray-Curtis score to represent the potential of interaction. This potential aims to measure how complementary are the signals that interacting cells produce. To do so, our Bray-Curtis score considers the number of active LR pairs that a pair of cells has while also incorporating the potential that each cell has to communicate independently (Fig 1). In other words, this score normalizes the number of active LR pairs used by a pair of cells by the total number of ligands and receptors that each cell expresses independently. Unlike other CCI scores that represent a directed relationship of cells by considering, for instance, only the number of ligands produced by one cell and the receptors of another, our CCI score is also undirected. To make our score undirected, it includes all ligands and receptors in cell A, and all cognate receptors and ligands, respectively, in cell B (Fig 1). Thus, pairs of cells interacting through all their ligands and receptors are represented by a value of 1 while those using none of them are assigned a value of 0.

## Genetic algorithm for selecting ligand-receptor pairs that maximize correlation between physical distances and CCI scores

An optimal correlation between intercellular distances and CCI scores was sought through a genetic algorithm (GA). This algorithm used as an objective function the absolute value of the Spearman correlation, computed after passing a list of ligand-receptor pairs to compute the CCI scores. In this case, only non-autocrine interactions were used (elements of the diagonal of the matrix with CCI scores were set to 0). The absolute value was considered because it could result either in a positive or negative correlation. A positive correlation would indicate that the ligand-receptor pairs used as inputs are preferably used by cells that are not close, while a negative value would indicate the opposite. The GA generated random subsets of the curated list of ligand-receptor pairs and used them as inputs to evaluate the objective function (as indicated in N2A Fig in S1 Text). The maximization process was run 100 times, generating 100 different lists that resulted in an optimal correlation. As shown in N2C-D Fig in S1 Text, a selection of the consensus ligand-receptor pairs was done according to their co-occurrence across the 100 runs of the GA and presence in most of the runs.

## Defining short-, mid- and long-range distances between cell types

The physical distances between all pairs of cell types in *C. elegans'* body were classified into different ranges of distances used for CCIs (short-, mid- or long-range distance) by using a Gaussian mixed model (N1 Fig in S1 Text). This model was implemented using the scikit-learn library for Python [71] and a number of components equal to 3.

## Benchmarking of CCI scores for representing intercellular distances

By using the same database of LR pairs, and same threshold of gene expression when pertinent, multiple CCI scoring methods were employed to compare the performance of our Bray-Curtis scoring approach:

I. The LR Count score was implemented by counting the number of active LR pairs in each cell-cell pair in an undirected manner (following the idea in Fig 1A).

II. We used the transcriptomic data and our LR database of *C. elegans* to run CellChat and compute the overall CCI weights between all cell-cell pairs among the 22 cell types with 3D coordinates of location. These results were exported into a matrix we called here A. To make this score undirected, we computed a matrix B = A + transpose(A). 1,000 permutations were used as a parameter of CellChat.

III. To implement the ICELLNET score, we computed the expression product of each LR pair using their log2(TPM+1) expression values as in [19], then the total sum was computed by considering both directions of interactions to make the final sum undirected.

IV. The score introduced in [33] (called here as Smillie score) was also implemented. Briefly, this score summarizes the overall likelihood of two cells to specifically interact, calculated as the -log10(P-value) resulting from 10,000 permutations. In each of these permutations, cell-type labels are shuffled in a way that the number of expressed ligands and receptors is preserved; then the strength of CCIs is computed. The strength in the original article is computed as the number of differentially expressed LR pairs in a cell-cell pair when comparing two conditions. However, we use only one condition, so the strength here simply corresponds to the number of active LR pairs in a cell-cell pair. The strength is recomputed every time that the cell-type labels are shuffled to build the null distribution. With the unshuffled strength and the null distribution, the P-value is computed as the probability to find values in the null distribution greater than the unshuffled strength. Finally the -log10(P-value) (Smillie score) is computed. To preserve the number of expressed ligands and receptors, we defined bins of size 10 and grouped cells together when they were in the same bin.

A Random Forest (RF) model was trained to predict ranges of intercellular distances (as defined in N1 Fig in S1 Text) from the CCI scores as inputs. This was done separately with each of the scoring methods (Bray-Curtis, CellChat, ICELLNET, LR Count, and Smillie scores) to measure the extent to which they can distinguish intercellular distances. For each method, the model training was performed using a stratified 3-fold cross-validation (CV). On each CV split a RF model with 10 estimators was trained and RF probability-predictions were compared to the test set using the Receiver Operating Characteristic (ROC). The Area Under the Curve (AUC) was computed for each CV split, and its mean and standard deviation was calculated across the CV splits. The RF classifier models were implemented through the XGBoost library for Python [72], and the performance evaluation including cross-validations and ROC curves was implemented through the Scikit-learn library for Python [71].

For the binary-based CCI scores (i.e., Bray-Curtis, LR Count and Smillie scores), further benchmarking of different gene-expression threshold values was done by training RF models as aforementioned. A value of > 10 TPM was selected as the threshold employed for all of these scores. For further discussion and details, see *Threshold values for the binary-based CCI scores* in S1 Text.

## Statistical analyses

For each function annotated in the list of ligand-receptor pairs (S1 Table), a one-sample Wilcoxon signed rank test was used to evaluate whether the relative abundance increased or decreased with respect to the distribution generated with the GA runs. In this case, a two-tailed test was performed for each function. Finally, the respective change was considered if the adjusted P-value passed the significance threshold (adj. P-value < 0.05).

A permutation analysis was done on the list of consensus ligand-receptor pairs obtained from the GA. To do so, three scenarios were considered: (1) a column-wise permutation (one column is for the ligands and the other for the receptors); (2) a label permutation (run

independently on the ligands and the receptors); and (3) a random subsampling from the original list, generating multiple subsets with similar size to the consensus list. In each of these scenarios, the list of ligand-receptor interactions was permuted 9,999 times.

All enrichment analyses in this work corresponded to a Fisher exact test. In all cases a P-value was obtained for assessing the enrichment and another for the depletion. The analysis of enriched ligand-receptor pairs along the body of *C. elegans* (head, mid-body and tail) was performed by considering all pairs of cells in each section and evaluating the number of those interactions that use each of the ligand-receptor pairs. The total number of pairs corresponded to the sum of cell pairs in all sections of the body. Similarly, the enrichment analysis performed for the different ranges of distance (short-, mid- and long-ranges) was done by considering all cell pairs in each range and the total number of pairs was the sum of the pairs in each range. To evaluate the enrichment of phenotypes (obtained through the tissue enrichment tool for *C. elegans* [52]), all genes in the GA-selected list were used as background. Then, the genes associated with the respective phenotype tested were used to assess the enrichment. For evaluating enrichment/depletion of ECM-component, membrane-bound, or secreted LR pairs in any of the intercellular-distance ranges, a Fisher exact test was used to compute the odds ratios and P-values by considering the active LR pairs across undirected cell-cell pairs: 1) those that were in both the type and the distance range, and 2) those that were not in in either of types or distance ranges.

When necessary, P-values were adjusted using Benjamini-Hochberg's procedure. In those cases, a significance threshold was set as FDR < 1% (or adj. P-value < 0.01).

## Selection of ligand-receptor pairs to analyze *in animal*

The ligand-receptor pairs selected for experimental validation of their gene expression had to met the following criteria: 1) the gene pairs have not been shown to interact in the cell types of interest, 2) they are expressed in only a few specific cell types (non-ubiquitous gene expression, based on Fig 4B), 3) just one of the gene pairs is highly expressed in one interacting cell and the other gene does so in the other interacting cell (to discard autocrine communication and evaluate interactions between LR pairs in different cell types) and 4) they are not highly expressed in cell types that are hard to differentiate under the microscope (e.g., GABAergic neurons are hard to distinguish from cholinergic neurons).

## *C. elegans* strains and husbandry

*C. elegans* PD4443 (ccIs4443[arg-1::GFP + dpy-20(+)]), KS411 (lin-17 (n671) I; unc-119 (e2498) III; him-5 (e1490) V; mhIs9[lin-17::GFP]), BC12925 (dpy-5 (e907) I; sIs10312 [rCesC05D11.4::GFP + pCeh361]), LX929 (vsIs48 [unc-17::GFP]), BC12890 (dpy-5 (e907) I; sIs11337[rCesY37A1B.5::GFP + pCeh361]) and PS3729 (unc-119 (ed4) III; syIs78[ajm-1::GFP + unc-119(+)]) strains were obtained from the Caenorhabditis Genome Center (CGC). For maintenance, the worms were typically grown at 20˚C on NGM plates seeded with *E. coli* strain OP50.

## Single molecule fluorescent *in-situ* hybridization

Single molecule fluorescent *in-situ* hybridization (smFISH) of L2 stage *C. elegans* was performed as previously described with some modifications [73]. Briefly, gravid worms of the strains of interest were bleached and the eggs rocked at 20˚C for 18 hours to synchronize the population. The L1 worms were then counted and around 5000 worms were seeded on NGM plates containing OP50 *E. coli* strain. Once the worms reached the L2 stage, they were harvested and then incubated in a fixation solution (3.7% formaldehyde in 1x PBS) for 45

minutes. The worms were then washed in 1x PBS and left in 70% ethanol overnight. The next day, the worms were incubated in wash buffer (10% formamide in 2x SSC) for 5 minutes before being incubated overnight at 30˚C in the hybridization solution containing the appropriate custom-made Stellaris FISH probes (Biosearch Technologies, United Kingdom). The samples were then washed twice in wash buffer for 30 minutes at 30˚C before being incubated in DAPI solution for nuclear counterstaining (10ng/mL in water) for 30 minutes at 30˚C. Finally, the stained worms were resuspended in 100µL 2x SSC and mounted on agar pads for fluorescent imaging on a Leica confocal microscope (Leica, Germany).

PD4443 worms expressing *arg-1*::GFP were incubated in probes targeting *lin-12* (CAL Fluor Red 590 dye) and GFP (Quasar 670 Dye), KS411 worms expressing *lin-17*::GFP were incubated in probes targeting *lin-44* (CAL Fluor Red 590 dye) and GFP (Quasar 670 Dye), and BC12925 worms expressing *let-756*::GFP were incubated in probes targeting *ver-1* (CAL Fluor Red 590 dye) and GFP (Quasar 670 Dye). Fluorescent imaging of GFP in PD4443 (*arg-1*), KS411 (*lin-17*) and BC12925 (*let-756*) was performed to ensure the expression patterns observed with smFISH were comparable (S8 Fig). Additionally, imaging of semo-1::GFP (BC12890) and ajm-1::GFP (PS3729) (S9 Fig), which have previously been used to define the location of the hypodermal cells [74], was performed to ensure correct annotation of the probe signal observed in smFISH. All images obtained from these conditions were analyzed and processed on Fiji [75].

## Supporting information

**S1 Text. Notes containing further details and discussion of particular points of the main manuscript.** It also includes N1-4 Fig.
(DOCX)

**S1 Table. Curated list of ligand-receptor interactions in *C. elegans*.**
(XLSX)

**S2 Table. Detailed information about ligand-receptor pairs that are used by pairs of cell types in *C. elegans*.**
(XLSX)

**S3 Table. Consensus list of ligand-receptor interactions selected by the genetic algorithm, corresponding to the "spatial code" of cell-cell interactions in *C. elegans*.**
(XLSX)

**S4 Table. Roles and experimental validation across literature of ligand-receptor pairs selected by the genetic algorithm.**
(XLSX)

**S5 Table. Ligand-receptor interactions of *C. elegans* described in literature.**
(XLSX)

**S6 Table. 3D digital atlas of *C. elegans* annotated with cell types in the RNA-seq data set.**
(XLSX)

**S1 Fig. UMAP visualization of CCIs using a Rand distance.** Visualization of the UMAP loadings computed for each pair of interacting cells. Dots represent pairs of interacting cells and they were projected based on their Rand distances (1-Rand index). In contrast to the Jaccard index that only accounts for true positives in the numerator, here the Rand index accounts for the true positives and negatives. It measures the number of agreements between two sets with respect to both the number of agreements and disagreements between these sets. Thus, the

Rand index in this case was computed as the number of active and inactive LR pairs present in both cell types simultaneously, and divided by the total number of LR pairs in the database used (245 in this case, S1 Table).
(TIFF)

**S2 Fig. Active pairs of ligand-receptor interactions across pairs of sender-receiver cells.**
Heatmap of presence or absence of ligand-receptor pairs (y-axis) across all combinations of sender-receiver cell types in C. elegans (x-axis). An agglomerative hierarchical clustering was performed on the Jaccard similarity for the ligand-receptor pairs (dendrogram for rows) and the pairs of cells (dendrogram for columns columns). Additionally, sender-receiver pairs were colored either by the sender cell or the receiver cell, according to the groups in the legend.
(TIFF)

**S3 Fig. Benchmarking of CCI scores to distinguish each of the intercellular distance ranges from the others.** Receiver operating characteristic (ROC) curves of random forest models for classifying cell-cell pairs from each of the CCI scores computed with different methods, as indicated in the legends. The classifiers predict the intercellular distance range (short-, mid-, or long-range distance, as defined in the N1C Fig in S1 Text). The performance is detailed through separate ROC curves for distinguishing each of the distance ranges from the rest using each of the CCI scores. For each classifier, the mean (solid line) ± standard deviation (transparent area) of the ROCs were computed with 3-fold stratified cross validations. The area under the curve (AUC) for the ROC curves is shown in the legend below, detailing the mean ± standard deviation from the cross-validations.
(TIFF)

**S4 Fig. Relative abundances of signaling functions across initial GA-LR pairs.** Composition plot given the signaling functions that LR pairs are associated with. Relative abundances are shown for the complete list of LR pairs (containing 245 interactions) and the subsets obtained in each of the 100 runs of the genetic algorithm (GA). Here, relative abundance is the number of LR pairs involved in a given pathway with respect to the total number of LR pairs in the list. Signaling functions are colored according to the legend.
(TIFF)

**S5 Fig. Active GA-LR pairs across undirected cell-cell pairs.** Heatmap of presence or absence of GA-LR pairs (y-axis) across all undirected cell-cell pairs in C. elegans (x-axis). Cell-cell pairs are sorted by their intercellular distances in an increasing manner, and are colored by the distance range as indicated above the colors (short-, mid-, and long-range distances, as defined in N1C Fig in S1 Text). Ligand-receptor interactions correspond to those in the list of GA-LR pairs, and each LR pair is considered present in an undirected cell-cell pair if it is used in either of the directed interactions between both cells. LR pairs are sorted and colored by the type of location where the ligand acts, as indicated to the left of the color (ECM-component, membrane-bound, or secreted).
(TIFF)

**S6 Fig. Expression of organ-phenotype associated genes in the LR pairs.** The presence or absence of proteins encoded by genes associated with organ system phenotype (y-axis) is indicated for each cell type (x-axis) according to C. elegans phenotype ontology. The threshold for presence is a gene expression value greater than 10 TPM; otherwise is labeled as absence. Only genes that are present in our complete list of LR pairs are shown, and members also in the GA-LR list are denoted with ochre cells (y-axis). Color keys for groups of cell types and GA-selection are depicted to the right. Agglomerative hierarchical clustering was performed using

a Jaccard similarity for both genes and cell types, independently.
(TIFF)

**S7 Fig. Comparison of cell-cell interaction scores used by the genetic algorithm to select ligand-receptor pairs.** Comparison of running our computational framework by using the Bray-Curtis CCI, LR Count, or ICELLNET scores. (A) Histogram of the maximal Spearman correlation achieved in 100 separate runs of the genetic algorithm when using these CCI scores. The colors in the legend indicate which score each distribution corresponds to. Dashed lines represent the median values in each distribution. As indicated to the right of the histograms, a Mann-Whitney U test was performed to compare the distributions in a pairwise manner. (B) Venn diagrams of the LR pairs present in the consensus list of LR pairs for each of the CCI scores, obtained from the 100 separate runs of the genetic algorithm in each case. The list indicated by the arrow shows the LR pairs that are contained in all consensus GA-LR pairs (intersection between GA-LR pairs of Bray-Curtis, LR Count and ICELLNET scoring methods).
(TIFF)

**S8 Fig. Validation of the expression patterns obtained by smFISH with GFP live imaging.** Expression patterns observed with smFISH overlap with those observed by live imaging of GFP, (A) *arg-1* expression in the rectal muscle in both smFISH and live imaging, (B) *let-756* expression in the non-seam hypodermal cells of the head in both smFISH and live imaging, (C) *lin-17* expression in the tail seam cells in both smFISH and live imaging. In all cases we changed the colors of the original images into magenta to make the visualizations comparable. Scale bar = 10μm.
(TIFF)

**S9 Fig. Confirmation of the localization of non-seam hypodermal cells expressing *lin-12* and *let-756*.** The expression patterns of *lin-12* in the tail (A) and *let-756* in the head (B) overlap with the expression patterns of *ajm-1* in the tail and *semo-1* in the head, confirming that the cells expressing *lin-12* and *let-756* in these regions correspond to non-seam hypodermal cells. Scale bar = 10μm.
(TIFF)

**S1 Movie. Tridimensional organization of cells expressing *arg-1* and *lin-12*.** Images of the smFISH analysis projected into the 3D space. The animation shows a rotation of this projection to reflect the 3D organization of cells. Here, the intestinal/rectal muscle and the non-seam hypodermal cells expressing *arg-1* (magenta) and *lin-12* (green), respectively, are shown in the tail of *C. elegans*, as indicated in Fig 6A.
(AVI)

**S2 Movie. Tridimensional organization of cells expressing *let-756* and *ver-1*.** Images of the smFISH analysis projected into the 3D space. The animation shows a rotation of this projection to reflect the 3D organization of cells. Here, the non-seam hypodermal and the amphid sheath cells expressing *let-756* (magenta) and *ver-1* (green), respectively, are shown in the head of *C. elegans*, as indicated in Fig 6B.
(AVI)

**S3 Movie. Tridimensional organization of cells expressing *lin-17* and *lin-44*.** Images of the smFISH analysis projected into the 3D space. The animation shows a rotation of this projection to reflect the 3D organization of cells. Here, the seam and the non-seam hypodermal cells expressing *lin-17* (magenta) and *lin-44* (green), respectively, are shown in the tail of *C. elegans*,

as indicated in Fig 6C.
(AVI)

## Acknowledgments

We thank Ariel Pani for helpful comments.

## Author Contributions

**Conceptualization:** Erick Armingol, Chintan J. Joshi, Eyleen J. O'Rourke, Nathan E. Lewis.

**Data curation:** Erick Armingol, Hratch Baghdassarian, Isaac Shamie, Samuel Berhanu, Anushka Dar, Fabiola Rodriguez-Armstrong, Olivia Yang.

**Formal analysis:** Erick Armingol, Abbas Ghaddar, Hratch Baghdassarian.

**Funding acquisition:** Eyleen J. O'Rourke, Nathan E. Lewis.

**Investigation:** Erick Armingol, Abbas Ghaddar.

**Methodology:** Erick Armingol, Abbas Ghaddar, Chintan J. Joshi, Hratch Baghdassarian.

**Project administration:** Erick Armingol.

**Software:** Erick Armingol, Chintan J. Joshi, Jason Chan, Hsuan-Lin Her.

**Supervision:** Eyleen J. O'Rourke, Nathan E. Lewis.

**Validation:** Erick Armingol, Abbas Ghaddar.

**Visualization:** Erick Armingol, Jason Chan.

**Writing – original draft:** Erick Armingol.

**Writing – review & editing:** Erick Armingol, Abbas Ghaddar, Hratch Baghdassarian, Eyleen J. O'Rourke, Nathan E. Lewis.

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
