## [Decision Letter · Decision Letter 0]

20 Jun 2022

Dear Dr. Lewis,

Thank you very much for submitting your manuscript "Inferring a spatial code of cell-cell interactions across a whole animal body" for consideration at PLOS Computational Biology.

As with all papers reviewed by the journal, your manuscript was reviewed by members of the editorial board and by several independent reviewers. In light of the reviews (below this email), we would like to invite the resubmission of a significantly-revised version that takes into account the reviewers' comments.

We cannot make any decision about publication until we have seen the revised manuscript and your response to the reviewers' comments. Your revised manuscript is also likely to be sent to reviewers for further evaluation.

Sincerely,

Pedro Mendes, PhD

Associate Editor

PLOS Computational Biology

Douglas Lauffenburger

Deputy Editor

PLOS Computational Biology

Reviewer's Responses to Questions

**Comments to the Authors:**

Reviewer #1: Deciphering the cell-cell interaction mechanisms for spatial organization and cellular functions is important. Here the authors presented a computational framework, cell2cell, to infer the spatial code of cell-cell interactions from scRNA-seq data. The key component of cell2cell lies in a newly defined cell-cell interaction score based on the coexpression of ligand-receptor pairs. An interesting part was the identification of a subset of ligand-receptor pairs that were strongly anti-correlated with distance via a genetic algorithm. The authors demonstrated the performance of cell2cell by leveraging a 3D altas of C.elegans’ cells and experimentally validating certain predictions. Overall, this study is very interesting and well suited for publication in PLoS Computational Biology. Below are a few comments that need to be addressed before publication.

1. The interaction score defined in this study was binary, which was different from the traditional methods that used the expression level of ligands and receptors. Can the authors provide a comparison analysis along with concrete examples to show the advantage of the binary communication score? Since many existing methods such as CellChat were based on the expression levels of ligands and receptors, it is very helpful if the authors can discuss the advantage and disadvantage of these two different scoring strategies.

2. The spatial transcriptomics is growing and there are many spatial transcriptomics datasets now. Due to the special body size of C. elegans, is the proposed method applicable to spatial transcriptomics of other tissues. The spatial location information from spatial transcriptomics data may be better to assess the correlations between the defined scores and the spatial distance.

3. The interaction score was defined for each pair of cells. A computational cost issue may raise when the number of cells was very large. For example, it may be challenging to project the cell-cell interactions onto a UMAP space like Fig. 2B.

Reviewer #2: In their manuscript “Inferring a spatial code of cell-cell interactions (CCIs) across a whole animal body”, the authors present a novel method for quantifying the degree to which cell types interact, inferred from their transcriptomes, then make the argument that the ligand-receptor (L-R) pairs that contribute to the relationship between intercellular interactions and their physical distance are involved in defining the tissue morphogenesis that drives that spatial arrangement. The major contributions of this work are the novel CCI scoring method, and that previously unknown roles in tissue organization can be inferred for L-R pairs that contribute to the relationship between interaction strength and physical proximity in cell type pairs.

The first contribution is of relevance to the field of single-cell transcriptomics, where it has become popular to infer CCIs from gene expression data. While this paper makes no novel contribution to the subgenre of that field interested in predicting the individual L-R pairs involved in these interactions, the modified Bray-Curtis dissimilarity this work proposes to quantify the relative “strength” of interactions between pairs of cell types is novel - albeit similar in goal (specificity of interaction) to Smillie et al. (https://doi.org/10.1016/j.cell.2019.06.029), where strength of CCIs were quantified as the probability of seeing as many L-R pairs between cell types by chance as calculated by a Monte Carlo simulation. No metric for CCI strength has been adequately evaluated experimentally in the current literature, so it is difficult to compare the author’s modified Bray-Curtis score to existing measures. However, the authors propose a novel evaluation metric by hypothesizing that interacting cells are in physical proximity, and thus CCI scores may be evaluated based on their relationship with physical distance between cell types. Taking advantage of the stereotypic arrangement of C. elegans cells and its recently published single-cell transcriptomic atlas, the authors are able to show that the modified Bray-Curtis score does reasonably well by this measure, showing a weak relationship between interaction strength and distance. I applaud the authors for this creative application of existing knowledge to validate their new metric. I would suggest that given their argument for the importance of a CCI score that captures specificity, as well as the similarity of their score to Smillie’s, the authors should consider using their metric to evaluate their score in the context of both Smillie’s statistical measure, and the commonly applied sum of L-R pairs as a CCI score. I’d expect their score to perform better than the sum of L-R pairs, and similar to Smillie’s. The advantage of the modified Bray-Curtis over Smillie’s statistical test is its simplicity, so as long as performance is similar, it would be fair to conclude that the Bray-Curtis score has an advantage.

My only concern is that the metric relies on accurately inferring L-R pairs which interact between cell types, something that is not a solved problem (I appreciate that the authors did note that these methods only have the ability to predict what L-R pairs could be used, not necessarily which are actually being used). Given that L-R interaction is inferred by expression over an arbitrary abundance threshold in this work, it would be interesting to see how changes in that threshold affect the correlation between CCI strength and physical distance. For example, if the expectation is that ligands expressed below threshold aren’t contributing to signaling, as the threshold is lowered the correlation may become weaker. Ultimately, the authors have proposed a novel cell-cell interaction prediction method cell2cell, and a creative solution to provide supporting evidence for its efficacy, and I’d like to see that used to demonstrate that the current parameterization of cell2cell is optimal, even if comparing cell2cell to other existing methods for inferring ligand-receptor interactions is beyond the scope of the paper.

The second contribution of this work is its method for hypothesis-generation of novel morphological roles for L-R pairs. With the rise of spatial scRNAseq, this could become a popular method for identifying candidate signals involved in tissue morphogenesis. My only question is whether the use of a complicated genetic algorithm was necessary in this case, as ultimately the spatial code was (at least in part) defined by relative enrichment of L-R pairs in each body section. Given that the genetic algorithm is the most computationally expensive part of the analysis, it would be interesting to see how skipping it (passing all L-R pairs to the spatial enrichment analysis) or using a less powerful but presumably quicker method to select L-R pairs driving the CCI-spatial correlation affects the efficiency of this method for generating accurate hypotheses regarding L-R pairs involved in morphogenesis. That being said, I appreciate that validating these hypotheses was not trivial, so consider this a suggestion for improving the general utility of the method rather than a requirement for publication.

The GA-LR result contains a consensus, but it would also be interesting to know how much redundancy there is in the set. Is there a lot of redundancy in general in the network? Could this be linked to robustness?

It is not clear what exactly the spatial code is and how strong it is. For example, is the spatial code just a correlation? And what use does this correlation have? Can we use it to improve LR interaction prediction? Or is it a skeleton physical network that matches animal anatomy? If the latter is the case, then you may be able to reconstruct aspects of the organism body plan based on the spatial code directly from scRNA-seq data. Is this possible? If we are to interpret the code directly as physical/biochemical, then additional questions may be raised that would be useful to explore to support the concept. For example, are cell adhesion (e.g. integrin, cadherin) relationships more likely to be expressed in nearby space compared to paracrine interactions?

The manual curation is suspicious and should be double-checked. e.g. F14B4.1 does not seem to be involved in CCI. It is labeled as a Wnt-receptor in table S1, but I can't find evidence of this in wormbase. There are a number of interactions with hsp-1 - are these all real CCIs? Is hsp-1 accessible to the exterior of the cell?

Minor notes:

The use of “spatial code” in the abstract / author summary is a little hard to follow as the concept hasn’t been adequately defined for the reader yet. By the end of the intro, the goal of the paper is clear, but in the abstract it might be necessary to spend a few more words than “spatial code” elaborating on the paper’s objective.

Fig 1: the visual attributes chosen are not consistently used. For example, the receptors have different colors (slightly, and difficult to differentiate) and the ligands don’t seem to. The shapes are different and that should be enough - it would be useful to simplify the shapes to make their complementarity easier to see. Another example is the purple to green gradient - it seems to be used for two things: one to distinguish the cells (that’s ok), and the other to highlight the difference between headings and data in the middle of the figure. Purple is further used for a few other shading areas that seem unrelated to the cell identity.

Fig2b: Jaccard score seems biased toward sender cells. While this does reflect previous findings and could be due to biology, Jaccard index can be biased in the case of class imbalance (because the numerator only considers true positives, ignoring true negatives) and the LR database contains twice the number of ligands as receptors. To defend against that critique, this analysis bears repeating with a different similarity metric better suited to imbalanced classes (perhaps adjusted Rand index).

Fig3a: Not very informative - the authors are trying to indicate changes in relative abundance of signaling functions per GA run output, but unfortunately it is difficult to follow. Perhaps 3a can be enlarged in the supplement for anyone skeptical about consistency between runs, and Fig3 can be a volcano plot of relative change (magnitude, x-axis) vs. significance (-log10 adj. p-value), which would be more effective at visually conveying the conclusion of Fig3b without getting bogged down in trying to represent each GA run.

Fig 4b: Perhaps matrix columns should match Fig4a for clarity?

“Thus, we hypothesized these PPIs represent a spatial code, that when used in different combinations can encode different spatial proximities.”

The combination claim is not supported

**Have the authors made all data and (if applicable) computational code underlying the findings in their manuscript fully available?**

Reviewer #1: Yes

Reviewer #2: Yes

PLOS authors have the option to publish the peer review history of their article (what does this mean?). If published, this will include your full peer review and any attached files.

Reviewer #1: **Yes: **Suoqin Jin

Reviewer #2: No
---

## [Decision Letter · Decision Letter 1]

7 Nov 2022

Dear Dr. Lewis,

We are pleased to inform you that your manuscript 'Inferring a spatial code of cell-cell interactions across a whole animal body' has been provisionally accepted for publication in PLOS Computational Biology.

Best regards,

Pedro Mendes, PhD

Academic Editor

PLOS Computational Biology

Douglas Lauffenburger

Section Editor

PLOS Computational Biology

Reviewer's Responses to Questions

**Comments to the Authors:**

Reviewer #1: The authors have well addressed my comments. Great work!

Reviewer #2: The authors have addressed all of the concerns of the previous review.

**Have the authors made all data and (if applicable) computational code underlying the findings in their manuscript fully available?**

Reviewer #1: Yes

Reviewer #2: Yes

PLOS authors have the option to publish the peer review history of their article (what does this mean?). If published, this will include your full peer review and any attached files.

Reviewer #1: No

Reviewer #2: No

---

## [Editor Report · Acceptance letter]

14 Nov 2022

PCOMPBIOL-D-22-00518R1 

Inferring a spatial code of cell-cell interactions across a whole animal body

Dear Dr Lewis,

I am pleased to inform you that your manuscript has been formally accepted for publication in PLOS Computational Biology. Your manuscript is now with our production department and you will be notified of the publication date in due course.

With kind regards,

Livia Horvath
